# Advance Time to Determine Injection and Flushing Times in Drip Fertigation

Gustavo Henrique da Silva *, Fernando França da Cunha  and Letícia Fonseca Anício de Brito

Department of Agricultural Engineering, Center of Agricultural Sciences, Federal University of Viçosa, Viçosa 36570-900, MG, Brazil
* Correspondence: gustavo.h.silva@ufv.br

**Abstract:** Drip fertigation has shown unquestionable benefits in recent decades compared to traditional farming fertilization practices. However, a fertilizer dissolved in the irrigation water must be evenly distributed in the fertigated area. Irrigation system and fertigation system characteristics and operational management potentially affect the uniformity of fertilizer and water distribution. Advance time (AT), which is an intrinsic and determinable characteristic of the irrigation system, has not been assessed as a useful technical criterion for managing the uniformity of fertilizer distribution in drip fertigation. The objective of this study was to assess the use of advance time as a technical criterion for determining the duration of injection time and flushing time that provides a satisfactory uniformity of spatial distribution of the fertilizer in drip fertigation. Therefore, the distribution uniformity of potassium chloride (KCl) fertilizer and water was evaluated at six injection times equivalent to 25, 50, 75, 100, 150, 200% AT, and two flushing times equivalent to 100 and 200% AT through Christiansen's uniformity coefficient (CU) and distribution uniformity (DU). The used drip irrigation system had 10 drip strips with 12.5 m length, a flow rate of 1.40 L h$^{-1}$ per dripper, and AT of 12.5 min. The injection solution was prepared with 40 g L$^{-1}$ of KCl. The results indicate that the distribution uniformity of KCl improved significantly with increasing injection time. The injection time of 200% AT promoted the greatest uniformity of distribution of KCl with CU of 0.977 and DU of 0.962. The flushing time of 100% AT was sufficient to rinse the irrigation system and promoted a satisfactory spatial distribution uniformity with a CU of 0.983. In both tests, the uniformity of irrigation water distribution was satisfactory, with CU of 0.988 and DU of 0.982 (average). Advance time is an intrinsic characteristic of the irrigation system that is useful in determining the duration of injection time and flushing time in a more technical way for drip fertigation with satisfactory spatial distribution uniformity of the fertilizer.

**Keywords:** advance time; distribution uniformity; fertilizer; Christiansen



## 1. Introduction

Fertigation consists in the application of water-soluble fertilizers to agricultural crops using irrigation water [1]. When properly employed, this technique can be practiced in any irrigation system. In drip irrigation, fertigation has shown unquestionable benefits in recent decades compared to traditional fertilizer application practices by farmers. Studies showed increases in crop yields, water productivity, and efficiency in the use of fertilizers [2,3], minimizing environmental damage by reducing N$_2$O emissions and nitrate leaching [4]. Drip fertigation is a promising technique for more sustainable food production. For this, the fertilizer solubilized in the irrigation water must be distributed evenly throughout the irrigated area.

An efficient fertigation event with high spatial distribution uniformity (SDU) of the fertilizer in the irrigation system area must have three timed and continuous intervals: (1) preinjection time: the period that starts with the activation of irrigation to fill the pipes with water until the complete stabilization of the flow and operating pressure of the

drippers, and to moisten the soil surface (and leaves); (2) injection time: the period of the injection of the solution containing the fertilizer into the irrigation system, which should be started after the stabilization of flow and pressure, and should rarely be less than 30 min; (3) flushing time: the period starting immediately after the end of injection time, which should last long enough to flush the pipe, remove the fertilizer from the leaves, incorporate the fertilizer into the root zone of the crop, and complete the irrigation depth [5,6].

In addition to the described times, several other factors, such as lateral layout [7,8], injector type [7,9,10], pressure [7,11], distance between injection point and main line [12], the length of the drip line [13], fertilizer concentration [8], injection time [14], and flushing time [15], potentially affect the uniformity of spatial distribution of water and fertilizers. However, the results of these studies show that the uniformity of fertilizer distribution improved when, directly or indirectly, the injection time or the flushing time were relatively long.

Variability in the characteristics of irrigation systems, such as layout, size, flow rate, pressure, spacing and drippers, hinders establish technical criteria. However, the advance time, which is an intrinsic and determinable characteristic of an irrigation system or sector, is a useful criterion in the management of fertigation in a centralized irrigation system with many sectors [16]. Advance time corresponds to the travel time that the fertilizer takes after being injected to reach a certain section or selected drip [6]. However, there are no studies that have examined the use of advance time to determine injection time and flushing time in drip irrigation, and their optimal duration to promote adequate fertilizer distribution uniformity.

The objective of this study was to assess the use of advance time as a technical criterion for determining the ideal duration of injection time and flushing time that provides satisfactory uniformity of the spatial distribution of the fertilizer in drip fertigation.

## 2. Material and Methods

### 2.1. Study Area and Drip Irrigation System

Experiments were carried out during October and November 2021 in a drip irrigation system (Figure 1) installed in a flat and grassy area located in the Experimental Area of Irrigation and Drainage belonging to the Agricultural Engineering Department of the Federal University of Viçosa, Viçosa-MG, Southeast Brazil (20°46′ S, 42°51′ W, 651 m). The drip irrigation system had ten drip tapes (Toro, Plentirain, China) with a length of 12.5 m, nominal diameter of 16 mm, and wall thickness of 0.6 mm, with adhesive-type inline drippers spaced every 30 cm (41 drippers per drip tape) with an average flow rate of 1.4 L h$^{-1}$ and operating pressure of 0.1 MPa maintained by a pressure-regulating valve. The drip tapes were spaced every 1.0 m and connected to a PVC pipe with nominal diameter of 50 mm. Irrigation water was stored in a 15 m$^3$ reservoir, and had pH of 6.7 and electrical conductivity of 68 μS cm$^{-1}$. A centrifugal pump was used to pressurize the irrigation system, and 120-mesh disc filters were used to prevent dripper clogging. Valves were installed at the ends of the tape and pipe for flushing after each fertigation to thoroughly rinse the pipe and avoid contamination of the following tests (Figure 1). More details of the layout are shown in Figure 1.

### 2.2. Fertilizer Injection and Advance Time

The injection solution was prepared with 40 g L$^{-1}$ of potassium chloride (KCl). The injection method was through the pressure difference using the negative pressure of the irrigation pump [17]. The injection flow rate was 60 L h$^{-1}$, equivalent to 10.5% of the irrigation system flow rate (574 L h$^{-1}$).

The advance time was determined for the 36th dripper among the 40 drippers of the 10th drip tape (Figure 1). In this irrigation system layout, this emitter is the farthest from the injection point. In practice, the section of the last five drippers is disregarded due to the low water velocity and the small area (large scale of layouts). The advance time was timed from the start of injection until the detection of salinity (first variation in EC value)

in the irrigation water discharged by the selected dripper through the EC meter (Lutron, CD-4303, Taipei, Taiwan). The advance time was 12.5 min.

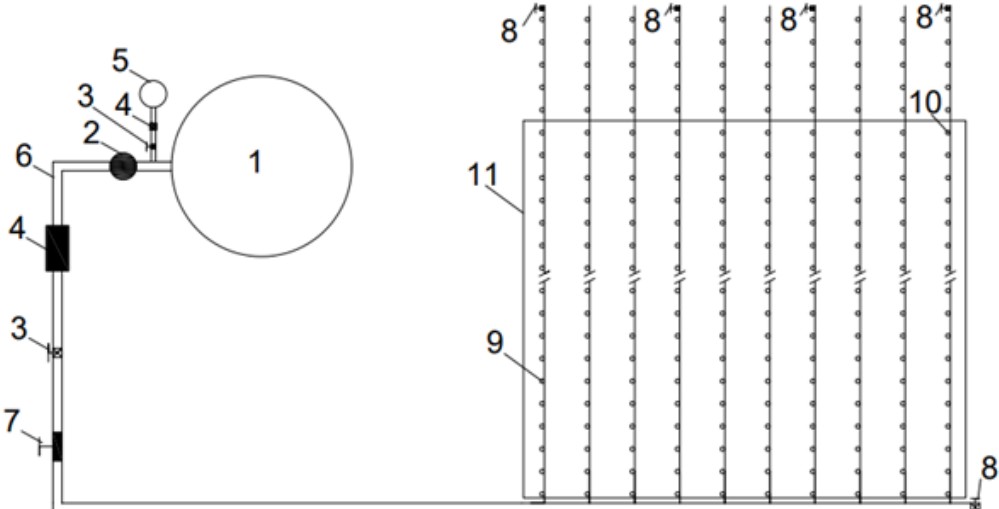

**Figure 1.** Schematic diagram of the drip irrigation system used to evaluate fertilizer and water spatial distribution uniformity. (1) Water reservoir; (2) pump; (3) valve; (4) filter; (5) fertilizer solution reservoir; (6) 50 mm diameter PVC pipe; (7) pressure-regulating valve; (8) flushing valve; (9) dripper; (10) ideal dripper to determine advance time in drip system; (11) monitored area.

### 2.3. Treatments

The treatments were injection time and flushing time, evaluated in two tests (Table 1). In Test 1, fertigation was performed with six injection times equivalent to 25, 50, 75, 100, 150 and 200% of the advance time (AT), adopting a preinjection time of 4.0 min and a flushing time of 200% AT. In Test 2, fertigation was performed with two flushing times equivalent to 100 and 200% AT, adopting a pressurization time of 4.0 min and an injection time of 200% AT. The experimental arrangement was simple, and the design was completely randomized with four replications. In a previous test, the EC value of the water discharged by the farthest dripper after the flushing time of 200% AT was equal to the EC value of the water from the reservoir, which indicates complete flushing of the pipes.

**Table 1.** Preinjection time (PIT), injection time (IT), and flushing time (FT) proportional to the advance time (AT), and fertigation duration (FD) in the two tests in drip fertigation. AT was 12.50 min.

| Tests | PIT | IT | FT | FD |
|---|---|---|---|---|
| | min | % AT (min) | % AT (min) | min |
| 1 | 4.0 | 200% (25.00) [1] | 200% (25.00) [1] | 54.0 |
| | 4.0 | 150% (18.75) | 200% (25.00) | 47.8 |
| | 4.0 | 100% (12.50) | 200% (25.00) | 41.5 |
| | 4.0 | 75% (9.38) | 200% (25.00) | 38.4 |
| | 4.0 | 50% (6.25) | 200% (25.00) | 35.3 |
| | 4.0 | 25% (3.13) | 200% (25.00) | 32.1 |
| 2 | 4.0 | 200% (25.00) | 100% (12.50) | 41.8 |
| | 4.0 | 200% (25.00) | 200% (25.00) | 54.0 |

[1] Values within parentheses in minutes.

### 2.4. Measurement of Fertigation Performance

To quantify the uniformity of spatial distribution of fertilizer and water, 24 collectors with volume of 4 L were systematically distributed in the area under six drippers (1st, 8th, 15th, 22nd, 29th and 36th drippers) in four drip tapes (1st, 4th, 7th, and 10th drip tapes) to collect the water discharged by the dripper. After each fertigation event, EC compensated at 25 °C (mS cm$^{-1}$) and the volume of discharged water (L) were measured in each collector

using a portable EC meter (LUTRON, CD 4303, Taipei, Taiwan) and a graduated cylinder, respectively. The concentration of KCl (g L$^{-1}$) was determined via the concentration versus EC curve compensated at 25 °C (Figure 2, Equation (1)). The quantity of KCl discharged by the dripper (g per dripper) was obtained by multiplying the concentration of KCl (g L$^{-1}$) by the volume of irrigation water collected (L).

$$EC_{25} = -0.0064 + 1.719\,C .\qquad(1)$$

where C is the KCl concentration in the solution (g L$^{-1}$) and EC$_{25}$ is the value of electrical conductivity compensated at 25 °C (mS cm$^{-1}$). The coefficient of determination of Equation (1) was 0.9992, which indicates a well-fitted equation that can be used to calculate the concentration of KCl fertilizer in the water discharged by the dripper.

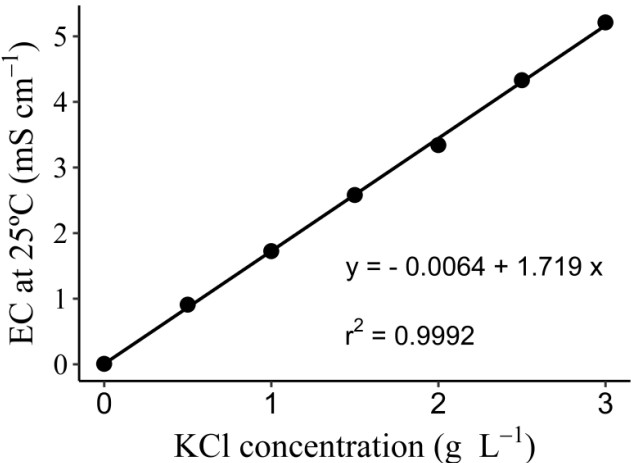

**Figure 2.** Relationship between electrical conductivity (EC) compensated at 25 °C and potassium chloride concentration (KCl) in the solution.

Fertilizer and water SDU in the irrigation system were determined using Christiansen's uniformity coefficient [18], described in Equation (2), and the distribution uniformity, described in Equation (3). These mathematical definitions are commonly used in studies to quantify uniformity in a system [7,8].

$$CU = 1 - \frac{\sum|X_i - X_m|}{\sum X_i}\qquad(2)$$

$$DU = \frac{X_{lq}}{X_m}\qquad(3)$$

where CU is Christiansen's uniformity coefficient, decimal; X$_i$ is the measure of the quantity of KCl (g) or the volume of water (L) in the i-th collector; X$_m$ is the mean value of the quantity of KCl (g) or the volume of water (L); DU is the distribution uniformity, decimal; X$_{lq}$ is the mean value of the quantity of KCl (g) or the volume of water (L) of the smallest quartile.

### 2.5. Data Analysis

The collected data were analyzed with analysis of variance and regression analysis. The regression model was selected on the basis of the significance of the regression coefficients by the *t*-test ($p < 0.01$), on the coefficient of determination (r$^2$) and on the biosystemic phenomenon. To perform the statistical analysis, R software was used [19].

Contour maps were prepared with the spatial distribution data of KCl (g per drip) and water (L per drip) using the "ggplot2" package of R software (version 4.1.0, Vienna, Austria) [19]. Because the total time of fertigation showed differences between the levels of

the treatments and consequently different amounts of KCl or water per drip, the values were standardized by dividing the i-th value by the average.

## 3. Results

### 3.1. Test 1—Uniformity of Spatial Distribution of KCL and Water as a Function of Injection Time

Injection time had no significant effect on water distribution, with mean CU of 0.988 and mean DU of 0.982 (Figure 3). The injection time had a significant effect on the uniformity of KCl distribution. CU and DU values decreased significantly with decreasing injection time. The longer injection time (200% AT) promoted satisfactory distribution uniformity, with CU of 0.977 and DU of 0.962, values very close to the values for irrigation water. The shortest injection time (25% AT) resulted in an unsatisfactory uniformity of KCl distribution, with a CU of 0.826 and a DU of 0.741. The variability in CU values (standard deviation) was lower for longer injection times (150 and 200% AT).

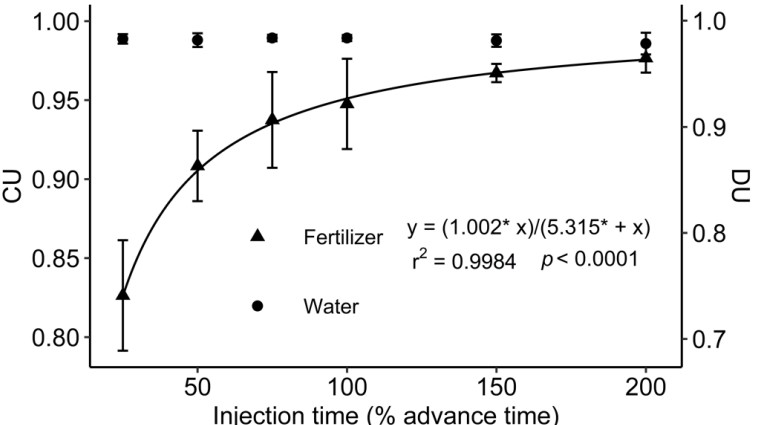

**Figure 3.** Christiansen's uniformity coefficient (CU) and distribution uniformity (DU) of fertilizer as a function of injection time (% of advance time). Error bar indicates the standard deviation for CU. * Significant coefficient via *t*-test ($p < 0.01$).

The injection time impacted the spatial distribution uniformity of KCL, according to the contour plots (Figure 4). In the shorter injection times (Figure 4A,B), the spatial distribution of KCl showed a deficit area in the final part of the initial drip lines (1st to 5th) and excess area in the final drip lines (7th to 10th). At the intermediate injection times (Figure 4C,D), the spatial distribution of KCl showed areas with deficit in the central and final parts of the first lateral lines (1st to 3rd), and relatively small areas with excess in the initial and final parts of the last ones (8th to 10th) in the fertigated area. At longer injection times (Figure 4E,F), the spatial distribution of KCl was more homogeneous with relatively small areas with deficit or excess. The spatial distribution of KCl with injection time equivalent to 200% AT showed homogeneity more similar to the spatial distribution of water. The spatial distribution of irrigation water was quite homogeneous (Figure 4G).

### 3.2. Test 2—KCl and Water Distribution Uniformity as a Function of Flushing Time

Flushing time had no significant effect on the uniformity of spatial distribution of KCl ($p = 0.29$) and water ($p = 0.27$) (Figure 5). In general, flushing times equivalent to 100% and 200% of the advance time were sufficient for flushing the irrigation system and promoted a satisfactory distribution uniformity, with CU of 0.983 and 0.977 ($p = 0.27$) for KCl and 0.990 and 0.986 ($p = 0.29$) for the irrigation water.

In general, the flushing times promoted satisfactory spatial distribution uniformity of KCl fertilizer and water, according to the contour plot (Figure 6). The spatial distribution of KCl showed a relatively small area with deficit at the end of the 10th lateral line for the 100% AT flushing time (Figure 6B) and an area with excess in the middle of the 10th lateral line for the 200% AT flushing time (Figure 6A). The spatial distribution of KCl with flushing

time equivalent to 200% AT showed more similar homogeneity to the spatial distribution of water. The spatial distribution of irrigation water was quite homogeneous (Figure 6B,D).

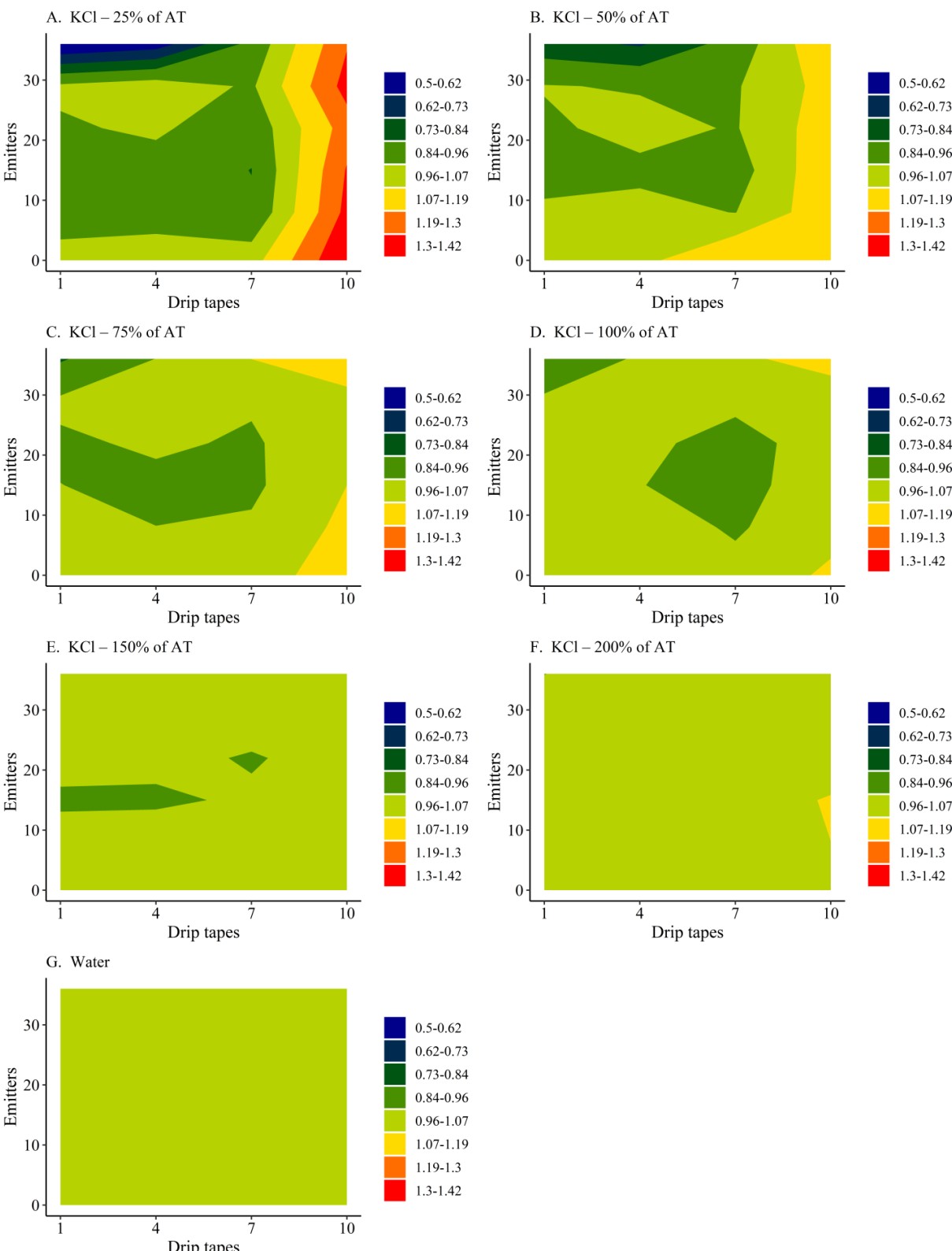

**Figure 4.** Spatial distribution of potassium chloride fertilizer for injection times of 25 (**A**), 50 (**B**), 75 (**C**), 100 (**D**), 150 (**E**) and 200 (**F**) % of advance time and of water (**G**). Transformed data ($X_i/\overline{X}$). Drip tape lines horizontally and drippers vertically.

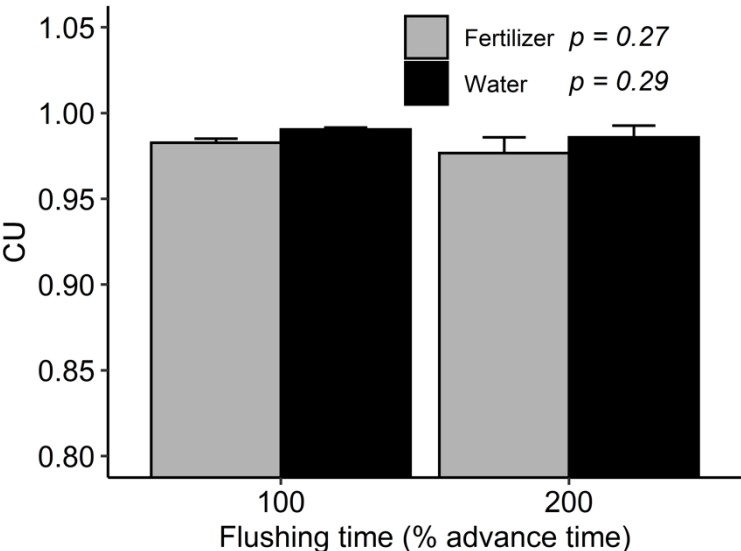

**Figure 5.** Christiansen's uniformity coefficient (CU) of KCl fertilizer and water for two flushing times (100 and 200% of advance time). *p*-values for the CU of fertilizer and water. Error bar indicates the standard deviation.

A. KCl – 100% of AT

B. Water

C. KCl – 200% of AT

D. Water

**Figure 6.** Spatial distribution of potassium chloride fertilizer and water for flushing times of 100 (**A,B**) and 200 (**C,D**) % of the advance time transformed data (Xi/$\overline{\text{X}}$). Drip tape lines, horizontal; dripper, vertical.

## 4. Discussion

The purpose of this study was to assess the impact of the injection time and flushing time, both calculated on the basis of advance time, and the uniformity of the spatial distribution of the fertilizer and drip fertigation water. Therefore, the spatial distribution uniformity of potassium chloride (KCl) and water was quantified using uniformity coefficients at six injection times (Test 1) and at two flushing times (Test 2).

The results of this study demonstrate that the advance time of the drip irrigation system was a useful technical criterion in determining the injection time and flushing time. The injection time equivalent to 200% of the advance time promoted a satisfactory KCl spatial distribution uniformity. Flushing times equivalent to 100% or 200% of the advance time promoted satisfactory KCl spatial distribution uniformity.

In fertigation, fertilizer distribution is conditioned to water distribution [7,8,10,12]. Therefore, the irrigation system must be properly designed to promote the high uniformity of water distribution with a DU greater than 0.90 [20]. The pressure and length of the drip tape are important factors in the sizing step, especially for drippers that do not compensate for pressure [21]. In the present study, the used irrigation system had favorable hydraulic conditions, such as a short lateral line length and large pipe diameter, which greatly reduced the head loss. Therefore, Christiansen's uniformity coefficient (CU) was 0.988, which is a very high value compared to those found under real field conditions [22].

The uniformity of spatial distribution of fertilizer and water is influenced by several characteristics of the fertigation system, irrigation system, and operational management [7,8,12,14]. Many characteristics hinder sizing and calibrating the fertigation system for each irrigation system (or sector). Therefore, the search for a main characteristic is essential to simplify the sizing and calibration of the fertigation system.

The time variable proved to be important in several studies on the uniformity of distribution in fertigation [14–16]. A uniform fertigation has three well-defined times: preinjection time, injection time, and flushing time [5]. Advance time, as an intrinsic and determinable feature of the irrigation system, can be useful in determining fertigation times for any drip irrigation system. The injection time equivalent to 200% of the advance time was ideal in this study. However, more research should be conducted with different characteristics of layout, manifold, flow, pressure, and size, and with other irrigation systems.

The advance time is defined as the travel time that the fertilizer, after being injected, takes to reach a certain section (or dripper) of the irrigation system [6]. In practice, the dripper farthest from the injection point should be chosen in each plot of the irrigation system, as the advance time is specific for each plot due the hydraulic characteristics. The advance time can be obtained by estimation based on the length, diameter, and flow of each section [6,16] or by in situ measurement, as performed in this study. The advance time starts with the injection of a solution with fertilizer or blue dye and ends with the detection of salinity with an EC meter or with the visualization of the bluish color, respectively, of the irrigation water.

The proposed method to determine the advance time in situ is suitable for a regular layout. Considering an irregular layout with different drip tape lengths, the critical dripper may have a different position than what was previously proposed. In the future, the proposed method needs to be tested in large-scale layouts with large numbers of plots, drip tapes, and drippers. Furthermore, conditions such as short injection time and low amount of fertilizer can hinder detecting electrical conductivity with an EC meter due the low concentration of fertilizers in the irrigation water. Therefore, these points must be taken into account in the correct measurement of the advance time in situ.

In practice, this study can be useful in assisting technicians and farmers in the sizing and calibration of the injector by defining the injection rate on the basis of the injection time and volume of the solution, which depends on the quantity and solubility of the fertilizer. It is worth mentioning that the type of injector can influence the uniformity of spatial distribution of the fertilizer. The ideal injector should have a constant injection rate and

provide constant concentration of the injected solution [7,10]. In this study, the constant injection rate and fixed KCl concentration in the solution were guaranteed.

The injection method by pressure difference was adopted in this study using the negative pressure or suction of the pump [17]. This injection method has some limitations. Fertilizer contact with the inside of the irrigation pump can cause corrosion and thus shorten its life. There are risks of water contamination at the point of capture in the event of backflow, which can cause serious environmental damage. Therefore, this method has restricted use and must be associated with the use of a check valve installed before the injection point.

A multi-sub-unit irrigation system with centralized fertigation can cause excess water when the total fertigation time is longer than the irrigation time needed to meet the crop's water demand [5]. In this case, a possible solution would be to move the injection point closer to the irrigation subunit [12], thus reducing the advance time, and consequently the injection and flushing times.

At the shortest injection time (25% AT), the spatial distribution of KCl was quite heterogeneous. This was a consequence of the rapid passage of irrigation water with the dissolved fertilizer through the submain line tube. This reduced the KCl entry into the initial drip strips, and consequently increased the KCl entry in the final drip strips, forming areas with deficit and excess.

After the injection time elapses, the fertilizer that dissolves in the irrigation water remains within the irrigation system. Therefore, additional operating time (flushing time) is required to rinse the pipes and thus discharge the fertilizer through the drippers. The permanence of fertilizers in the pipes can cause clogging of the drippers, which reduces the distribution uniformity of the irrigation system [23]. In this study, flushing time equal to or greater than the advance time was sufficient, corroborating other studies [6,24].

Understanding the minimal fertigation time (injection time plus flushing time) is important to define the timing of injection during the irrigation time as a function of soil nutrient dynamics. Fertigation carried out in the second half of the irrigation time incorporates less fertilizer into the soil profile, which promoted greater loss by nitrogen (ammonia) volatilization [25]; however, on the other hand, it may reduce the downward movement of nitrate [25], potassium, and sulfur [26], minimizing the leaching of these nutrients and consequently reducing groundwater contamination.

The findings of this study can help technicians and farmers in more accurately defining the injection and flushing times from the advance time for fertigation in a drip irrigation system to ensure the satisfactory uniformity of the spatial distribution of the fertilizer.

## 5. Conclusions

Advance time is a useful technical criterion for determining the duration of injection time and flushing time to achieve the satisfactory spatial distribution uniformity of the fertilizer in drip fertigation.

**Author Contributions:** Conceptualization, G.H.d.S. and F.F.d.C.; methodology, L.F.A.d.B. and G.H.d.S.; software, G.H.d.S.; validation, G.H.d.S., F.F.d.C. and L.F.A.d.B.; formal analysis, G.H.d.S. and F.F.d.C.; investigation, G.H.d.S.; resources, G.H.d.S. and F.F.d.C.; data curation, L.F.A.d.B.; writing—original draft preparation, G.H.d.S.; writing—review and editing, L.F.A.d.B. and F.F.d.C.; visualization, L.F.A.d.B.; supervision, G.H.d.S. and F.F.d.C.; project administration, F.F.d.C.; funding acquisition, G.H.d.S. and F.F.d.C. All authors have read and agreed to the published version of the manuscript.

**Funding:** Brazilian National Council for Scientific and Technological Development (CNPq), scholarship grant 164662/2018-9, and Coordination for the Improvement of Higher Education Personnel (Capes), finance code 001.

**Institutional Review Board Statement:** Not applicable.

**Informed Consent Statement:** Not applicable.

**Data Availability Statement:** Available upon request.

**Acknowledgments:** The authors are grateful to the Brazilian National Council for Scientific and Technological Development (CNPq), scholarship grant 164662/2018-9, and the Coordination for the Improvement of Higher Education Personnel (Capes), finance code 001.

**Conflicts of Interest:** The authors declare no conflict of interest.

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
