# Peer review of "Advance Time to Determine Injection and Flushing Times in Drip Fertigation"

_horticulturae, doi:10.3390/horticulturae8121103_

Round 1

Reviewer 1 Report

This work had evaluated the use of advance time as a technical criterion for determining the duration of injection time and flushing time with a spatial distribution uniformity of drip fertigation. The methodology used in the present study is sound and the findings interesting. In my opinion, this manuscript is well-written and provides a new insight for determining the injection and flushing time of drip fertigation. I just have a few points that need to be addressed:

1. Line 78-81: Is the length of each drip tape 12.0 m?

2. Line 97-98How to get the irrigation system flow rate?

3. Line 99-101Why should be the 36th drippier of the 10th drip tape chose? How much were the water velocities of the 36th drippier of the 10th drip tape and the last five drippers respectively?

4. Line 101-104: Is the travel time the advance time? Please clarify the expression.

5. Line 193: Are the legends P=0,27 and 0,29 or 0.27 and 0.29 in Figure 5?

6. Line 296-299: Does the conclusion of the manuscript correspond to the title?

Author Response

Resposta aos comentários do revisor 1

Ponto 1: Este trabalho avaliou o uso do tempo de avanço como critério técnico para determinar a duração do tempo de injeção e o tempo de descarga com uniformidade de distribuição espacial da fertirrigação por gotejamento. A metodologia utilizada no presente estudo é sólida e os resultados interessantes. Na minha opinião, este manuscrito está bem escrito e fornece uma nova visão para determinar a injeção e o tempo de irrigação da fertirrigação por gotejamento. Eu só tenho alguns pontos que precisam ser abordados:

Linha 78-81: O comprimento de cada fita gotejadora é 12,0 m?

Resposta 1: Sim. Cada fita gotejadora tem 12,5 m. Os 12,0 m foram substituídos por 12,5 m.

Ponto 2: Linha 97-98: Como obter a vazão do sistema de irrigação?

Resposta 2: A vazão dos emissores foi determinada em teste de campo, utilizando coletores (catlatas) para coletar a água descarregada pelo emissor. A vazão do sistema era a vazão do emissor multiplicada pelo número total de emissores.

Ponto 3: Linha 99-101: Por que o 36º drippier da 10ª fita de gotejamento deve ser escolhido? Qual foi a velocidade da água do 36º gotejador da 10ª fita gotejadora e dos últimos cinco gotejadores, respectivamente?

Resposta 3: o 36º emissor da 10ª fita gotejadora é o mais distante do ponto de injeção. Para água + adubo chegar neste emissor é preciso passar pelo maior comprimento da tubulação + fita gotejadora.

As velocidades da água do 36-31º gotejador da 10ª fita gotejadora calculadas usando comprimento, diâmetro e vazão foram 0,0106, 0,0088, 0,0069, 0,0050, 0,0031, 0,0013 m/s. O tempo de avanço calculado foi de mais de 6 minutos no último comprimento da fita gotejadora.

Ponto 4: Linha 101-104: O tempo de viagem é o tempo de avanço? Por favor, esclareça a expressão.

Resposta 4: As expressões foram usadas com o mesmo significado no manuscrito. Para padronização, os autores utilizarão o tempo de antecedência.

Ponto 5: Linha 193: As legendas P=0,27 e 0,29 ou 0,27 e 0,29 na Figura 5?

Resposta 5: P=0,27 para UC de fertilizante ep=0,29 para UC de água. Os autores modificam no manuscrito para maior clareza. E “,” foi substituído por “.” na Figura 5.

Ponto 6: Linha 296-299: A conclusão do manuscrito corresponde ao título?

Resposta 6: Os autores revisaram o título e a conclusão e modificaram para “ADVANCE TIME TO DETERMINE INJECTION AND FLUS

Reviewer 2 Report

Uniform distribution of irrigation and fertilizer is a very important technique, and I think this paper to be of great technical value.

This experiment was conducted on an experimental farm and is small in scale. I think that an empirical study on a larger scale, on actual production plots, is needed. If you have already conducted such a large-scale experiment, please add it to this paper. If you have not yet done so, report the results of your experiments in the future.

Fertigation methods which include small amounts of water at a time and frequent irrigation(fertigation) are reported to improve the efficiency of fertilizer use. I would like you to have a discussion on the design of irrigation systems, suitable for such fertigation methods, if possible.

Ikeda, H. (2007). Environment-friendly soilless culture and fertigation technique. Acta Hortic. 761, 363-369
DOI: 10.17660/ActaHortic.2007.761.50
https://doi.org/10.17660/ActaHortic.2007.761.50

Line 99:

You performed the measurement of the advance time on the 36th dripper of the 10th drip tape. You need to explain the reason or rule for why you conducted the measurement here. The reader may wonder where the advance time should be measured in the case of large plots with a large number of drip tapes, a large number of drippers, and a large scale of plots.

Author Response

Response to Reviewer 2 Comments

Point 1: Uniform distribution of irrigation and fertilizer is a very important technique, and I think this paper to be of great technical value.

This experiment was conducted on an experimental farm and is small in scale. I think that an empirical study on a larger scale, on actual production plots, is needed. If you have already conducted such a large-scale experiment, please add it to this paper. If you have not yet done so, report the results of your experiments in the future.

Response: The authors have not tested the proposed method on large-scale layouts. But the necessary experiments of the future were reported in the Discussion.

Fertigation methods which include small amounts of water at a time and frequent irrigation(fertigation) are reported to improve the efficiency of fertilizer use. I would like you to have a discussion on the design of irrigation systems, suitable for such fertigation methods, if possible.

Ikeda, H. (2007). Environment-friendly soilless culture and fertigation technique. Acta Hortic. 761, 363-369
DOI: 10.17660/ActaHortic.2007.761.50
https://doi.org/10.17660/ActaHortic.2007.761.50

Response: In this condition addressed by you, the injection of fertilizers must be continuous because the irrigations are daily. So that all irrigation has fertigation as is soilless cultivation (substrate) and greenhouse.

However, there are field conditions such as clay soil where irrigation has low frequency (irrigation intervals). In this case, the fertilizer residual inside the pipeline must be removed during the flushing time after each fertigation.

Some characteristics of dripper spacing, nutrient soil mobility, wet bulb, irrigation frequency, soil texture, type of crop, weather, water quality influence the characteristics of project and management of irrigation and fertigation system, the discussion must be deeper. The authors beliefs that should be at another time.

Line 99:

You performed the measurement of the advance time on the 36th dripper of the 10th drip tape. You need to explain the reason or rule for why you conducted the measurement here. The reader may wonder where the advance time should be measured in the case of large plots with a large number of drip tapes, a large number of drippers, and a large scale of plots.

Response: In practice, the dripper farthest from the injection point should be chosen in each plot of the irrigation system, as the advance time will be specific for each plot due the hydraulic characteristics.

The authors rewrote the lines 117-120 and 271-274 for clarity.
